# Dopamine increases protein synthesis in hippocampal neurons enabling dopamine-dependent LTP

**Tanja Fuchsberger[1]\*, Imogen Stockwell[2], Matty Woods[1], Zuzanna Brzosko[1], Ingo H Greger[2], Ole Paulsen[1]\***

[1]Department of Physiology, Development and Neuroscience, Physiological Laboratory, University of Cambridge, Cambridge, United Kingdom; [2]Neurobiology Division, MRC Laboratory of Molecular Biology, Cambridge, United Kingdom

**\*For correspondence:**
tf337@cam.ac.uk (TF);
op210@cam.ac.uk (OP)

**Competing interest:** The authors declare that no competing interests exist.

## eLife Assessment

This manuscript addresses a mechanism by which dopamine (DA) regulates synaptic plasticity. The authors build upon their previous finding that DA applied after a timing pattern that ordinarily induces long-term depression (LTD) now induces long-term potentiation (LTP). The new findings that this 'DA-dependent LTP' involves de novo protein synthesis, a cyclicAMP signalling pathway, and calcium-permeable AMPA receptors (CP-AMPARs) are of **valuable** significance. The conclusions are **convincing** and largely supported by the evidence provided.

**Abstract** The reward and novelty-related neuromodulator dopamine plays an important role in hippocampal long-term memory, which is thought to involve protein-synthesis-dependent synaptic plasticity. However, the direct effects of dopamine on protein synthesis, and the functional implications of newly synthesised proteins for synaptic plasticity, have not yet been investigated. We have previously reported that timing-dependent synaptic depression (t-LTD) can be converted into potentiation by dopamine application during synaptic stimulation (Brzosko et al., 2015) or postsynaptic burst activation (Fuchsberger et al., 2022). Here, we show that dopamine increases protein synthesis in mouse hippocampal CA1 neurons, enabling dopamine-dependent long-term potentiation (DA-LTP), which is mediated via the $Ca^{2+}$-sensitive adenylate cyclase (AC) subtypes 1/8, cAMP, and cAMP-dependent protein kinase (PKA). We found that neuronal activity is required for the dopamine-induced increase in protein synthesis. Furthermore, dopamine induced a protein-synthesis-dependent increase in the AMPA receptor subunit GluA1, but not GluA2. We found that DA-LTP is absent in GluA1 knock-out mice and that it requires calcium-permeable AMPA receptors. Taken together, our results suggest that dopamine together with neuronal activity controls synthesis of plasticity-related proteins, including GluA1, which enable DA-LTP via a signalling pathway distinct from that of conventional LTP.

## Introduction

An important role of dopamine in hippocampal long-term memory has long been recognised (*O'Carroll et al., 2006*; *Rossato et al., 2009*; *Shohamy and Adcock, 2010*; *Takeuchi et al., 2016*). Long-term memory is thought to be encoded by synaptic plasticity, in particular long-term potentiation (LTP; *Bliss and Lomo, 1973*) which, based on its duration, has been divided into an early phase and a protein-synthesis-dependent late phase (*Frey et al., 1988*). Dopaminergic signalling has been implicated specifically in 'late-phase' LTP (L-LTP; *Frey et al., 1990*; *Huang and Kandel, 1995*; *Matthies*

*et al., 1997*). Dopamine acts on G-protein-coupled receptors which activate adenylate cyclases (AC) to generate cAMP, which in turn activates several effectors, including the cAMP-dependent protein kinase (PKA). PKA has several downstream targets which ultimately regulate transcription and translation (*Sassone-Corsi, 1995*; *Kebabian and Calne, 1979*; *Mayr and Montminy, 2001*; *Smith et al., 2005*). Although the requirement of dopamine for protein-synthesis-dependent LTP has been well established, the underlying cellular mechanisms of how dopamine modulates activated synapses via protein-synthesis-dependent mechanisms remain poorly understood.

In order to selectively label newly synthesised proteins, we used a puromycin-based assay (*Schmidt et al., 2009*), adapted for use in acute mouse hippocampal slices. We found that dopamine increases protein synthesis in hippocampal neurons. We then investigated whether protein synthesis is required for a recently described dopamine-dependent form of plasticity (dopamine-dependent long-term potentiation [DA-LTP]) which, although being rapid in onset, otherwise shares properties with L-LTP (*Brzosko et al., 2015*; *Fuchsberger et al., 2022*). Whilst conventional 'early' LTP induced by a spike-timing-dependent plasticity protocol (t-LTP; *Bi and Poo, 1998*) was unaffected, DA-LTP was completely abolished by either of two different protein synthesis inhibitors.

A previous study in primary cultured hippocampal neurons reported that dopaminergic signalling increases the surface expression of the AMPA receptor subunit GluA1 (*Smith et al., 2005*), which has been widely studied in the context of LTP (*Huganir and Nicoll, 2013*; *Diering and Huganir, 2018*). Although most hippocampal AMPA receptors are heteromeric, incorporating the edited form of the GluA2 subunit, rendering the receptor calcium-impermeable, GluA1 may form homomeric calcium-permeable AMPA receptors (CP-AMPARs; *Burnashev et al., 1992*; *Wenthold et al., 1996*). As well as enabling calcium-permeability of the AMPAR, by lacking GluA2, the extracellular N-terminal domain of GluA1 in GluA1 homomers is highly flexible, which alters the gating of the receptor and hinders its synaptic anchoring (*Zhang et al., 2023*; *Stockwell et al., 2024*). CP-AMPARs have been implicated in some forms of LTP (*Plant et al., 2006*; *Guire et al., 2008*; *Purkey and Dell'Acqua, 2020*), including a PKA-dependent form of plasticity (*Park et al., 2021*).

We found that levels of the GluA1 receptor subunit, but not GluA2, were upregulated in response to dopamine in a protein-synthesis-dependent manner. Moreover, DA-LTP was absent in a GluA1 genetic knock-out (KO) mouse model, while conventional t-LTP remained intact. These findings suggest that newly synthesised GluA1 receptor subunits mediate the expression of DA-LTP, possibly by forming GluA1 homomeric CP-AMPARs. Indeed, while blocking CP-AMPARs did not interfere with conventional t-LTP, they were required for DA-LTP.

## Results

### Dopamine increases protein synthesis in CA1, which is required for DA-LTP

We hypothesised that the application of dopamine induces synthesis of proteins that are required for the expression of DA-LTP. To address this, we first investigated whether dopamine directly affects protein synthesis in hippocampal neurons. In order to exclusively visualise newly synthesised proteins in hippocampal slices, we used a puromycin (PMY)-based assay (*Schmidt et al., 2009*). Labelling was achieved by incubating acute hippocampal slices in artificial cerebrospinal fluid (aCSF) containing 3 µM PMY for 30 min (*Figure 1A*). As expected, slices incubated with PMY showed a significantly higher PMY intensity than those without (control, 0.39±0.02 a.u., n=9, vs +PMY, 0.54±0.03, t(16) = 4.0, p=0.001, n=9; *Figure 1B and C*), confirming specificity of the antibody labelling approach. Furthermore, we tested whether PMY incorporation and labelling are specific to protein synthesis by adding the protein synthesis inhibitor cycloheximide (CHX) or anisomycin (AM) to the slices. We found that 30 min preincubation with 10 µM CHX significantly reduced the PMY signal, confirming the specificity of the labelling approach (PMY, 1.8±0.45, n=7 vs PMY + CHX, 0.4±0.05, n=8, t(6) = 3.1, p=0.02; *Figure 1—figure supplement 1A and B*), as did 2 hr preincubation with 0.5 mM AM (PMY, 0.6±0.14, n=4 vs PMY + AM, 0.16±0.03, n=5, t(3.2)=3.1, p=0.048; *Figure 1—figure supplement 1C and D*). All results are reported as mean ± SEM.

Strikingly, when applying 100 µM dopamine, the PMY signal was significantly increased compared with slices without dopamine (PMY, 0.54±0.03, n=9 vs PMY + DA, 0.80±0.019, n=16, p<0.001; *Figure 1B and C*). We next examined the effect of the D1/D5 receptor agonist SKF38393 (10 µM) and

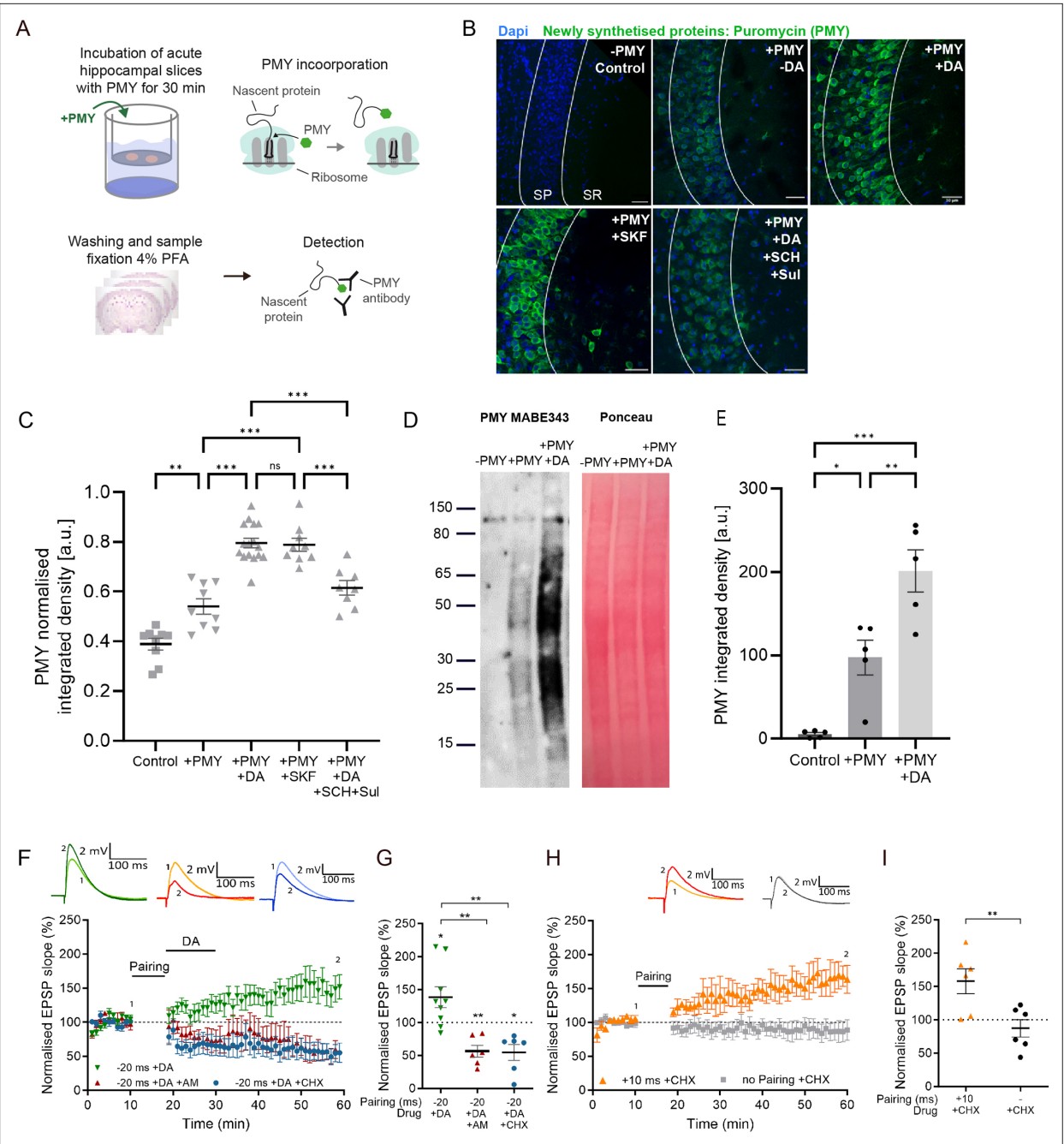

**Figure 1.** Dopamine increases protein synthesis in CA1, required for dopamine-dependent long-term potentiation (DA-LTP), but not for conventional t-LTP. (**A**) Experimental workflow for protein synthesis labelling in acute hippocampal slices using a puromycilation assay. PMY=puromycin. (**B**) Representative images of the CA1 region of the hippocampus (SP: stratum pyramidale, SR: stratum radiatum) of puromycin-labelled proteins (green) and Dapi (blue) in the following conditions: negative control (no PMY), PMY alone, PMY +dopamine (DA), PMY+SKF38393 (SKF), and PMY+DA +SCH23390 (SCH)+sulpiride (Sul). The slices show an increase in protein synthesis after DA or SKF38393 application, which is blocked by DA receptor antagonists SCH23390 and sulpiride. Scale bar, 30 µm. (**C**) Summary of results. One-way ANOVA followed by Tukey's HSD test, **, $p < 0.01$; ***, $p < 0.001$; ns, not significant. (**D**) Representative western blot of newly synthesised proteins detected by PMY Mabe343 antibody shows no signal in negative control (-PMY), and increased PMY signal in the presence of dopamine (+DA). Ponceau stain confirms equal loading of total protein. (**E**) Summary of results. One-way ANOVA followed by Tukey's HSD test, *, $p < 0.05$; **, $p < 0.01$; ***, $p < 0.001$. (**F**) Dopamine application (DA) after a post-before-pre pairing protocol (Pairing, $\Delta t = -20$ms) leads to synaptic potentiation (green trace), which is blocked by postsynaptically applied anisomycin (AM; red trace) or cycloheximide (CHX; blue trace). (**G**) Summary of results. t-test, **, $p < 0.01$. (**H**) Intact conventional t-LTP ($\Delta t = +10$ms, orange trace) in the presence of CHX. (**I**) Summary of results. t-test, **, $p < 0.01$. Traces show an excitatory postsynaptic potential (EPSP) before (1) and 40 min after (2) pairing. Plots show averages of normalised EPSP slopes ± SEM.

*Figure 1 continued on next page*

*Figure 1 continued*

The online version of this article includes the following source data and figure supplement(s) for figure 1:

**Source data 1.** Normalised EPSP slopes of all recorded cells and data points of statistics figures.

**Source data 2.** Original file of the full raw uncropped, unedited western blot.

**Source data 3.** Uncropped western blot with the relevant bands labelled.

**Figure supplement 1.** Puromycin (PMY) assay is specific to protein synthesis and protein synthesis inhibitors do not affect baseline synaptic transmission within 60 min of recording.

**Figure supplement 1—source data 1.** Normalised EPSP slopes of all recorded cells and data points of statistics figures.

found that it stimulates protein synthesis to a similar extent as dopamine, with no significant difference between those two conditions (PMY + DA, 0.80±0.019, n=16, vs PMY + SFK, 0.79±0.026, n=9, p>0.99), and significantly different to PMY alone (PMY, 0.54±0.03, n=9 vs PMY + SFK, 0.79±0.026, n=9, p<0.001; *Figure 1B and C*). We next tested whether the dopamine-induced increase in protein synthesis could be prevented by the application of D1/5 and D2 receptor antagonists. Using 10 μM SCH23390, a selective antagonist at D1 (Ki = 0.2 nM) and D5 (Ki = 0.3 nM) receptor subtypes, and 50 μM sulpiride, a D2 antagonist (Ki = 8 nM), we found that the dopamine-induced increase in PMY signal was prevented (PMY + DA, 0.80±0.019, n=16, vs PMY + DA + SCH23390/sulpiride, 0.61±0.029, n=8, p<0.001; all p values reported above from post-hoc Tukey's HSD test, after confirming a statistically significant difference between at least two groups in one-way ANOVA; F(4, 46)=49.55, p<0.001; *Figure 1B and C*).

To confirm the effect of dopamine on protein synthesis with an alternative quantification method, acute hippocampal slices were incubated as above and homogenates prepared for western blot analysis of the micro-dissected CA1 region. As expected, without PMY incubation, the signal was virtually absent compared with samples with PMY (neg control, 5±2, n=5, vs +PMY, 97±21, n=5, p=0.01). Dopamine application significantly increased the PMY signal (PMY, 97±21, n=5, vs PMY + DA = 201±25, n=5, p=0.006; p values reported from post-hoc Tukey's HSD test after one-way ANOVA; F(3, 16)=19.85, p<0.0001; *Figure 1D and E*).

We then tested whether newly synthesised proteins are required for DA-LTP induced by a spike-timing-dependent long-term depression (t-LTD) pairing protocol followed by dopamine application during low-frequency afferent stimulation, which converts synaptic depression into potentiation (*Brzosko et al., 2015*). Conventional t-LTD induced by a post-before-pre pairing without dopamine application leads to synaptic depression, whereas a pre-before-post t-LTP protocol induces synaptic potentiation (*Bi and Poo, 1998*; *Feldman, 2012*). We confirmed that the application of dopamine after the post-before-pre pairing protocol (Δt = -20ms) induces robust synaptic potentiation (138% ± 16% vs 100%, t(8) = 2.5, p=0.039, n=9; *Figure 1F and G*), while the same pairing protocol without dopamine application leads to robust input-specific synaptic depression (t-LTD 64% ± 9% vs control 106% ± 8%, t(6) = 5.8, p=0.001; *Figure 1—figure supplement 1E and F*). Dopamine application alone had no effect on synapses that did not undergo prior pairing (102% ± 14% vs 100%, t(8) = 0.13, p=0.89, n=9; *Figure 1—figure supplement 1G*). When delivering the protein synthesis inhibitor AM to the postsynaptic cell through the recording pipette, DA-LTP was fully blocked, leaving synaptic depression instead (DA + AM, 56% ± 9% vs 100%, t(5) = 4.8, p=0.0047, n=6), which was significantly different from dopamine without AM (DA 138% ± 16% vs DA + AM 56% ± 9%, t(13) = 3.9, p=0.0016; *Figure 1F and G*). To exclude the possibility that the effect of AM on blocking DA-LTP is mediated via other signalling pathways affected by AM (*Croons et al., 2009*; *Hazzalin et al., 1998*; *Kyriakis et al., 1994*) rather than by specific inhibition of protein synthesis, we used CHX, an alternative blocker of protein synthesis. We found that loading CHX into the postsynaptic cell through the recording pipette also completely blocked DA-LTP, leaving synaptic depression instead (DA + CHX, 55% ± 12% vs 100%, t(5) = 3.7, p=0.013, n=6; *Figure 1F*), which was significantly different from dopamine without CHX (DA 138% ± 16% vs DA + CHX 55% ± 12%, t(13) = 3.9, p=0.002; *Figure 1F and G*). We also confirmed that postsynaptically applied AM or CHX had no effect on baseline synaptic transmission throughout the 60-min duration of the experiment in the control pathway (no pairing +AM, 104±17% vs 100%, t(5) = 0.23, p=0.83, n=6; and no pairing +CHX, 98±6% vs 100%, t(5) = 0.32, p=0.76, n=6; *Figure 1—figure supplement 1H and I*).

In contrast, when we used a t-LTP protocol (Δt = +10ms) while CHX was loaded into the postsynaptic cell, we still observed robust potentiation (t-LTP +CHX, 158% ± 19% vs 100%, t(5) = 3.11, p=0.026, n=6, *Figure 1H*), and, as expected, there was no significant effect of CHX on the control pathway (CHX, 87% ± 13% vs 100%, t(5) = 0.97, p=0.37, n=6, *Figure 1H*). These results demonstrate that DA-LTP, but not conventional t-LTP, requires postsynaptic protein synthesis, revealing two different signalling pathways for LTP, one of which requires dopamine and protein synthesis, while the other one does not.

## Dopamine and neuronal activity mediate increase in protein synthesis

DA-LTP requires neuronal activation during dopamine application either via subthreshold synaptic stimulation (*Brzosko et al., 2015*) or postsynaptic bursts (*Fuchsberger et al., 2022*). Thus, we investigated whether neuronal activity is also required for the dopamine-induced increase in protein synthesis.

To test this, acute hippocampal slices were treated with 1 µM tetrodotoxin (TTX), a voltage-gated sodium channel blocker, during PMY incubations (*Figure 2A*). We confirmed using whole-cell patch clamp recordings that application of TTX inhibits spontaneous excitatory postsynaptic potentials (EPSPs) in CA1 pyramidal neurons in our preparations (*Figure 2B*). Using the PMY labelling assay, we found that TTX incubation alone did not significantly alter PMY labelling, suggesting that the baseline level of protein synthesis is not affected by blocking neuronal activity for 30 min (control, 0.55±0.033, n=8, vs TTX, 0.53±0.026, n=8, p=0.90; *Figure 2A and C*). However, TTX significantly diminished the dopamine-induced increase in protein synthesis (DA, 0.83±0.024, n=9, vs DA +TTX, 0.65±0.017, n=9, p<0.001). Nevertheless, the dopamine-induced increase in protein synthesis was not fully blocked by TTX, as it remained significantly higher in the presence of dopamine and TTX compared with TTX alone (TTX +DA, 0.65±0.017, n=9, vs TTX 0.53±0.026, n=8, p=0.005; p values reported from post-hoc Tukey's HSD test after one-way ANOVA; F (3, 30)=30.57, p<0.001; *Figure 2A and C*).

## Dopamine-induced increase in protein synthesis is mediated via AC and PKA

We then sought to identify the signalling pathway that mediates the changes in protein synthesis in response to dopamine and neuronal activity. D1 and D5 dopamine receptors are known to activate ACs, which increase cyclic adenosine monophosphate (cAMP) activating the cAMP-dependent PKA (*Sassone-Corsi, 2012*). We used the PMY incubation assay to assess the protein synthesis level in the presence of Rp-cAMPS, a cell-permeable cAMP analogue which acts as an inhibitor of PKA. We found that the application of 30 µM Rp-cAMPS had no significant effect on protein synthesis in control conditions (control, 1.2±0.10, n=10, vs Rp-cAMPS 1.7±0.4, n=8, p=0.25) but completely blocked the dopamine-induced increase in PMY levels (DA, 2.2±0.2, n=10, vs DA +Rp-cAMPS, 1.3±0.1, n=9, p=0.029; p values reported from post-hoc Tukey's HSD test after one-way ANOVA; F (3, 33)=4.912, p=0.0062; *Figure 2D and E*).

AC subtypes 1 and 8 are additionally activated by $Ca^{2+}$, which would make them attractive candidates to mediate activity-dependent dopamine signalling. We therefore tested whether DA-LTP requires these ACs using an AC1/AC8 double knock-out (DKO) mouse model (*Wang et al., 2003*). We found that DA-LTP was completely absent in AC1/AC8 DKO mice, leaving synaptic depression instead (AC DKO 74% ± 10% vs 100%, t(6) = 2.6, p=0.04, n=7), which was significantly different to DA-LTP in WT mice (WT 125% ± 16%, n=6, vs AC DKO 74% ± 10%, n=7, t(11) = 4.2, p=0.0016; *Figure 2F and H*). To test whether PKA signalling is required for DA-LTP, we loaded 1 µM of the PKA blocker, protein kinase inhibitor-(6-22)-amide (PKI), into the postsynaptic cell through the recording pipette. Dopamine application after the t-LTD priming protocol followed by subthreshold synaptic stimulation failed to convert depression into potentiation in the presence of PKI (DA + PKI 81% ± 3% vs 100%, t(5) = 6.1, p=0.0017, n=6), which was significantly different from the effect of dopamine application without PKI (DA + PKI 81% ± 3% vs DA 125% ± 16%, t(10) = 6.2, p=0.0001; *Figure 2G and H*).

## Dopamine increases GluA1, which is required for DA-LTP

It has been reported in hippocampal primary neuronal culture that dopaminergic stimulation enhances surface expression of the AMPA receptor subunit GluA1 (*Smith et al., 2005*). To confirm whether this finding holds in an acute slice preparation, we tested the effect of dopamine on GluA1 levels

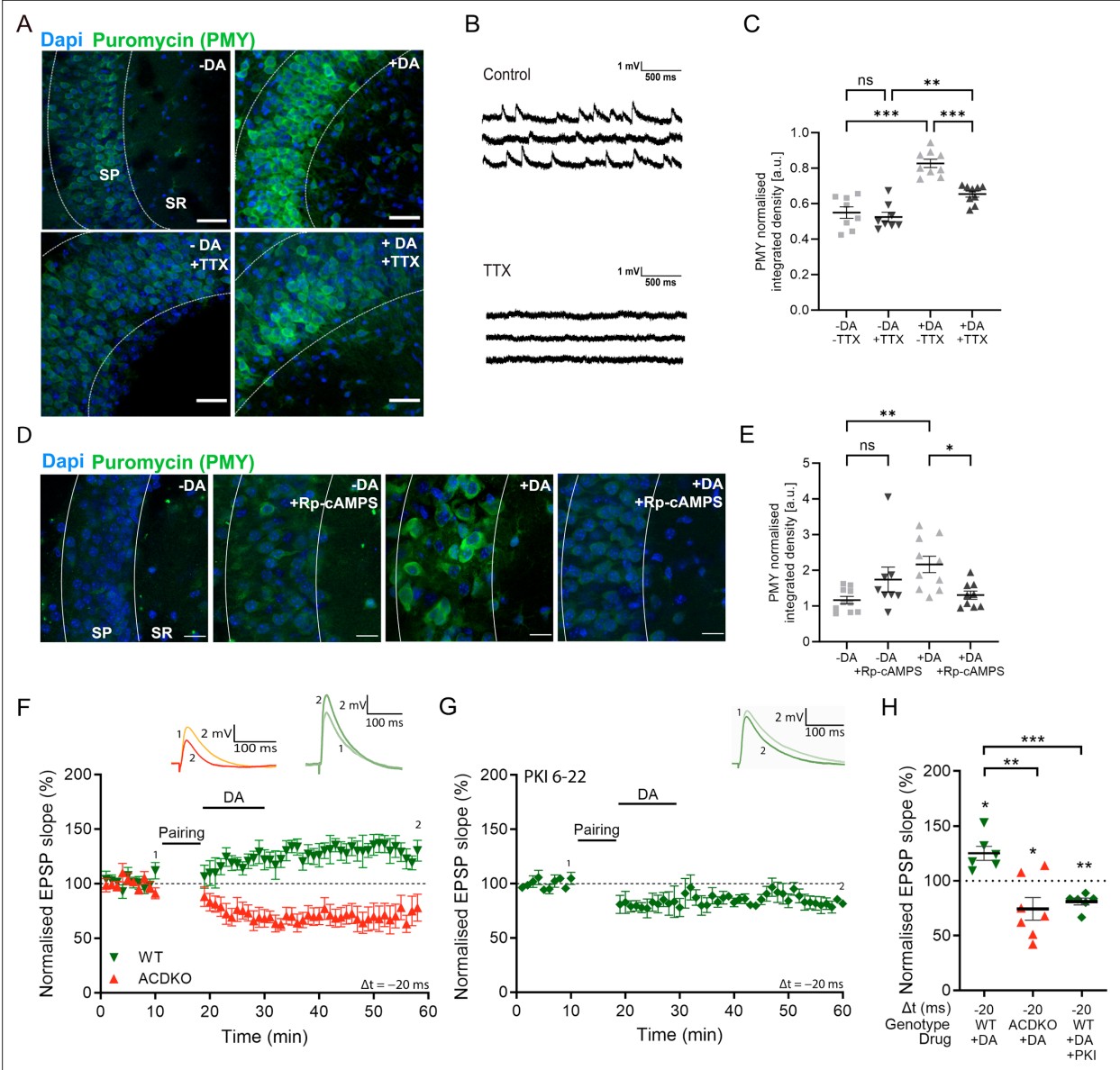

**Figure 2.** Dopamine and neuronal activity mediate increase in protein synthesis via AC1/8 and PKA enabling dopamine-dependent long-term potentiation (DA-LTP). (**A**) Representative images of the CA1 region (SR: stratum oriens, SP: stratum pyramidale) of the hippocampus of puromycin (PMY)-labelled proteins in the following conditions: PMY alone (-DA), PMY +dopamine (+DA), PMY + TTX (-DA+TTX), PMY + dopamine + TTX (+DA+TTX). Images show that the dopamine-induced increase in protein synthesis is reduced in the presence of TTX. Scale bar, 30 µm. (**B**) TTX abolishes spontaneous activity shown in traces of whole-cell patch clamp recording. (**C**) Summary of results. One-way ANOVA followed by Tukey's HSD test, **, p<0.01; ***, p<0.001; ns, not significant. (**D**) Representative images of the CA1 region (SR: stratum oriens, SP: stratum pyramidale) of the hippocampus of PMY-labelled proteins in the following conditions: PMY alone (-DA), PMY +Rp-cAMPS (-DA+Rp-cAMPS), PMY +dopamine (+DA), and PMY + dopamine + Rp-cAMPS (+DA + Rp-cAMPS). Images show that the dopamine-induced increase in protein synthesis is prevented by Rp-cAMPS. (**E**) Summary of results. One-way ANOVA followed by Tukey's HSD test, *, p<0.05; **, p<0.01; ns, not significant. (**F**) Dopamine application (DA) after a post-before-pre pairing protocol (Pairing, Δt = -20ms) leads to synaptic potentiation in WT (green trace), but not in AC DKO mice (red trace). (**G**) Postsynaptically applied PKA inhibitor PKI6-22 blocks DA-LTP (green trace), leaving synaptic depression instead. (**H**) Summary of results. One-way ANOVA followed by Tukey's HSD test, **, p<0.01; ***, p<0.001. Traces show an excitatory postsynaptic potential (EPSP) before (1) and 40 min after (2) pairing. Plots show averages of normalised EPSP slopes ± SEM.

The online version of this article includes the following source data for figure 2:

**Source data 1.** Normalised EPSP slopes of all recorded cells and data points of statistics figures.

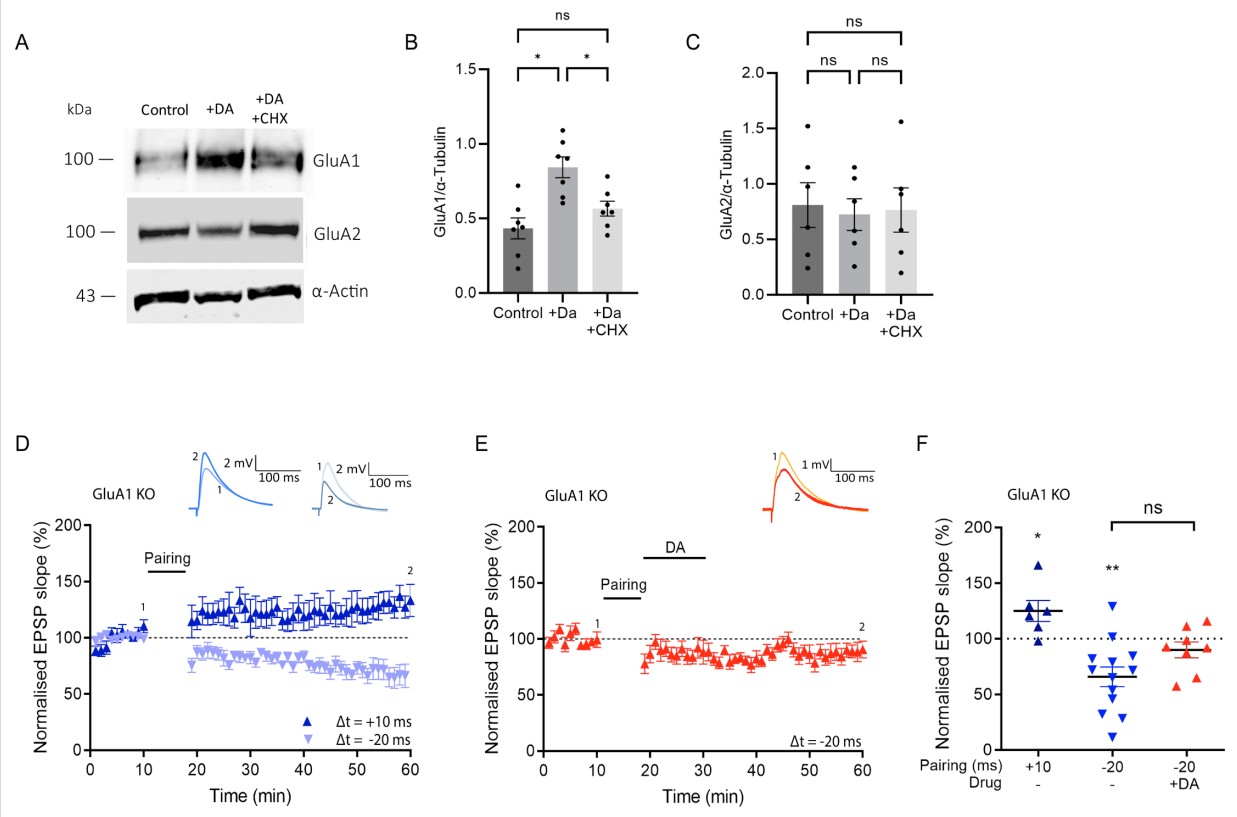

**Figure 3.** Dopamine increases GluA1 in a protein-synthesis-dependent manner, which is required for dopamine-dependent long-term potentiation (DA-LTP). (**A**) Western blot images from tissue homogenates of the hippocampal CA1 region show increase in GluA1 upon dopamine application (+DA), which is abolished in the presence of cycloheximide (+DA + CHX). α-Actin was used as loading control. Western blot shows unchanged GluA2 following dopamine application (DA) and no change with cycloheximide (+DA + CHX). (**B, C**) Summary of results. One-way ANOVA followed by Tukey's HSD test, *, p<0.05; ns, not significant. (**D**) A t-LTP pairing protocol (Δt = +10 ms) induces potentiation (dark blue trace), and a t-LTD protocol (Δt = -20 ms) induces depression (light blue trace) in GluA1 KO mice, (**E**) No DA-LTP in GluA1 KO mice. (**F**) Summary of results. All traces show an excitatory postsynaptic potential (EPSP) before (1) and 40 min after (2) pairing. Plots show averages of normalised EPSP slopes ± SEM. t-test, *, p<0.05; **, p<0.01; ns, not significant.

The online version of this article includes the following source data and figure supplement(s) for figure 3:

**Source data 1.** Normalised EPSP slopes of all recorded cells and data points of statistics figures.

**Source data 2.** Original files of the full raw uncropped, unedited western blots.

**Source data 3.** Uncropped western blot with the relevant bands labelled.

**Figure supplement 1.** Whole blot images of western blots presented in main *Figure 3*.

in the micro-dissected CA1 region. Using western blot analysis, we found that dopamine application induces a significant increase in the GluA1 receptor subunit compared with control conditions (control, 0.43±0.069 a.u., n=7, vs DA, 0.84±0.07, n=7, $F_{(1.8, 10.8)}$=12.04, p=0.013; *Figure 3A and B*). Importantly, when we blocked protein synthesis with CHX, the GluA1 increase was prevented (DA, 0.84±0.07, n=7, vs DA + CHX, 0.38±0.05, n=7, $F_{(1.8, 10.8)}$=12.04, p=0.037; *Figure 3A and B*), showing that the dopamine-induced increase in GluA1 is protein-synthesis-dependent. In contrast, when we measured the levels of the GluA2 subunit under the same conditions, we could not detect significant differences between dopamine-treated and control CA1 (control, 0.8±0.2, n=6, vs DA, 0.7±0.14, n=6, $F_{(1.4, 6.88)}$=0.86, p=0.46; *Figure 3A and C*), nor in the presence of CHX (DA 0.7±0.14, n=6, vs DA + CHX, 0.76±0.19, n=6, $F_{(1.4, 6.88)}$=0.86, p=0.87; *Figure 3A and C*). Whole western blot images are shown in *Figure 3—figure supplement 1*.

We next investigated the functional implications of GluA1 for DA-LTP using a GluA1 KO mouse model. GluA1 plays an important role in synaptic plasticity (*Zamanillo et al., 1999*; *Granger et al., 2013*; *Park et al., 2019*; *Purkey and Dell'Acqua, 2020*) but is not required for all forms of LTP

(*Hoffman et al., 2002*; *Romberg et al., 2009*; *Frey et al., 2009*). Thus, we first tested whether we could induce conventional t-LTP and t-LTD in hippocampal slices from GluA1 KO mice. We found that a t-LTP protocol (Δt = +10 ms) led to synaptic potentiation, albeit of somewhat reduced magnitude (t-LTP GluA1 KO, 125% ± 9% vs 100%, t(5) = 2.6, p=0.046, n=6; *Figure 3D and F*), while a t-LTD protocol (Δt = −20 ms) induced synaptic depression (t-LTD GluA1 KO, 66% ± 9% vs 100%, t(12) = 3.9, p=0.002, n=13; *Figure 3D and F*). Strikingly, however, in the GluA1 KO mice, the application of dopamine failed to convert t-LTD into LTP (DA-LTP GluA1 KO, 90% ± 7%, n=8; *Figure 3E*), and the resulting depression was not significantly different from t-LTD without dopamine (t-LTD vs DA-LTP, t(19) = 1.905, p=0.07; *Figure 3F*). Taken together, these results suggest that newly synthesised GluA1 subunits are required for the expression of DA-LTP.

## DA-LTP requires CP-AMPARs

GluA1 homomers may form CP-AMPARs in the hippocampus (*Wenthold et al., 1996*). A transient increase in CP-AMPARs has been reported in some forms of LTP (*Plant et al., 2006*; *Guire et al.,*

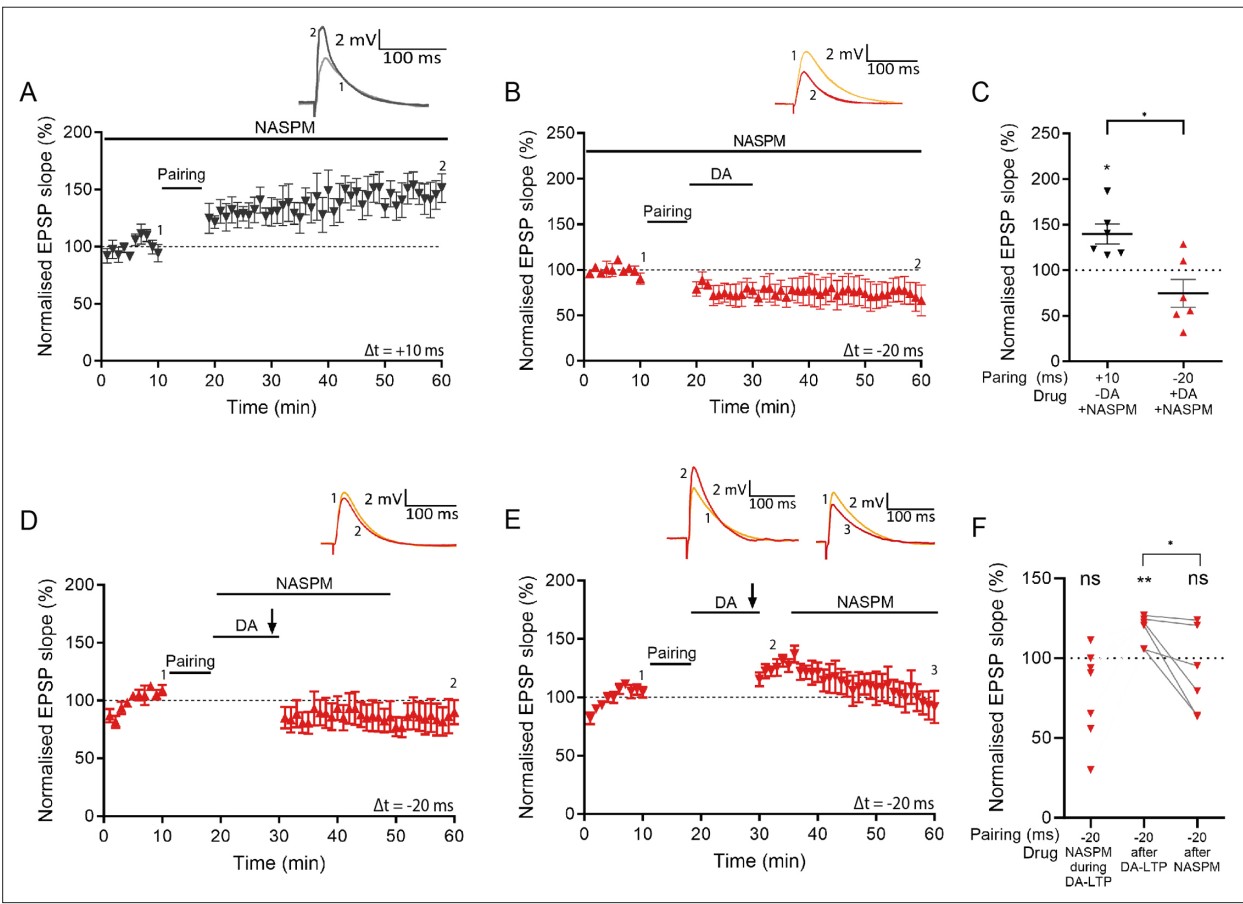

**Figure 4.** CP-AMPARs are required for dopamine-dependent long-term potentiation (DA-LTP) but not for conventional t-LTP. (**A**) A t-LTP pairing protocol (Δt = +10 ms) induces synaptic potentiation in the presence of extracellularly applied 1-naphthyl acetyl spermine (NASPM). (**B**) NASPM blocks DA-LTP. (**C**) Summary of results. *t*-test, *, p<0.05. (**D**) Burst-induced DA-LTP is blocked by NASPM. (**E**) Burst-induced DA-LTP potentiation decreases when NASPM is applied 7 min afterwards. (**F**) Summary of results. *t*-test, *, p<0.05; **, p<0.01; ns, not significant. Traces in (**A**, **B**, and **D**) show an excitatory postsynaptic potential (EPSP) before (1) and 40 min after pairing (2). Traces in (**E**) show an EPSP before (1), 5 min after DA and burst stimulation (2), and 40 min after pairing (3). Plots show averages of normalised EPSP slopes ± SEM.

The online version of this article includes the following source data and figure supplement(s) for figure 4:

**Source data 1.** Normalised EPSP slopes of all recorded cells and data points of statistics figures.

**Figure supplement 1.** CP-AMPARs blocker IEM-1460 confirms that CP-AMPARs are required for dopamine-dependent long-term potentiation (DA-LTP) but not for conventional t-LTP.

**Figure supplement 1—source data 1.** Normalised EPSP slopes of all recorded cells and data points of statistics figures.

*2008*; *Park et al., 2019*; *Purkey and Dell'Acqua, 2020*), including a PKA-dependent form of plasticity (*Park et al., 2021*). Since we observed a protein-synthesis-dependent increase in the GluA1, but not GluA2, AMPA receptor subunit after dopamine application, we hypothesised that CP-AMPA receptors might be involved in DA-LTP.

We compared the effect of 1-naphthyl acetyl spermine (NASPM; 100 µM), a selective CP-AMPAR antagonist, on conventional t-LTP (Δt = +10 ms) to the effect of NASPM on DA-LTP. We found that robust t-LTP was elicited in the presence of extracellular NASPM (139% ± 11% vs 100%, t(5) = 3.6, p=0.015, n=6; *Figure 4A and C*). In contrast, when applying NASPM during a DA-LTP protocol, potentiation was completely prevented (75% ± 15% vs 100%, t(5) = 1.66, p=0.16, n=6; *Figure 4B and C*). This shows that CP-AMPARs are indeed required for DA-LTP. The selective increase in GluA1 receptor subunit levels seen in response to dopamine (*Figure 3A–D*) suggests that GluA1-containing CP-AMPARs are required for the expression of DA-LTP.

We have recently reported that, in addition to synaptic activation, postsynaptic burst stimulation can also induce DA-LTP, which is mediated via the same AC-PKA signalling pathway and also requires protein synthesis (*Fuchsberger et al., 2022*). Synaptically induced DA-LTP develops gradually, while burst-induced DA-LTP shows rapid potentiation. We therefore used burst-induced DA-LTP to test whether a transient increase in CP-AMPARs may be required for potentiation. We applied NASPM during or shortly after the plasticity protocol to test whether this affects the expression of DA-LTP. We found that the application of NASPM during burst stimulation completely prevented synaptic potentiation (78% ± 11% vs 100%, t(6) = 2.01, p=0.091, n=7; *Figure 4D and F*). When applying NASPM 7 min after burst stimulation, we observed an initial potentiation (118% ± 4% vs 100%, t(6) = 4.52, p=0.0063, n=6), which gradually returned to baseline in the presence of NASPM (91% ± 11% vs 100%, t(5) = 0.82, p=0.45, n=6; *Figure 4E and F*), suggesting that the expression of DA-LTP is mediated by CP-AMPARs.

To confirm these results with an alternative CP-AMPAR blocker, we applied 10 µM IEM1460 extracellularly. Conventional t-LTP (Δt = +10 ms) was compared with DA-LTP in the presence of IEM1460 throughout the recording, which were significantly different (t-LTP +IEM1460, 142%±21%, n=7, vs DA-LTP +IEM1460, 80%±8%, n=7, t(12) = 2.75, p=0.02; *Figure 4—figure supplement 1A, B and D*). When applied immediately after the pairing protocol, IEM1460 still prevented DA-LTP and left a synaptic depression instead (DA-LTP +IEM after, 68% ± 12% vs 100%, t(8) = 2.68, p=0.028, n=9; *Figure 4—figure supplement 1C and D*). Taken together, these results suggest that DA-LTP, but not conventional t-LTP, requires CP-AMPARs for the expression of synaptic potentiation.

## Discussion

In summary, we investigated the effect of dopamine on protein synthesis in hippocampal neurons and how protein synthesis enables DA-LTP. We report four main findings: (1) Dopamine increases protein synthesis in an activity-dependent manner through the activation of PKA. (2) Dopamine enables a rapid onset protein-synthesis-dependent form of synaptic potentiation (DA-LTP). (3) Dopamine increases the level of GluA1 but not GluA2 subunit of AMPA receptors. (4) The expression of DA-LTP requires GluA1 AMPA receptor subunit and $Ca^{2+}$-permeable (CP)-AMPARs, whereas t-LTP does not, suggesting the existence of two distinct forms of LTP.

Recent developments of protein synthesis labelling techniques enabled us to directly monitor protein synthesis in neurons in response to dopamine. We validated our PMY-based approach for labelling protein synthesis using protein synthesis inhibitors AM and CHX in acute hippocampal slices. We report that dopamine increases protein synthesis in pyramidal neurons in CA1 of acute hippocampal slices within minutes. This is consistent with previous reports from hippocampal and cortical neuronal cell culture systems, which showed that dopamine receptor agonist SKF-38393 enhances protein synthesis (*Smith et al., 2005*; *David et al., 2020*). We found that the dopamine-induced increase in protein synthesis is mediated by dopamine receptors via the AC-cAMP-PKA pathway (*Mayr and Montminy, 2001*; *Sassone-Corsi, 1995*), which is also consistent with previously reported hippocampal cell culture results which showed that the application of Sp-cAMPS, an activator of PKA, was sufficient to induce an increase in protein synthesis (*Smith et al., 2005*). In addition, the AC-cAMP-PKA pathway induced by dopamine may affect several downstream targets and interact with other signalling pathways that could enhance protein synthesis. It has been reported in cortical

primary neuronal cultures that D1 receptors, but not D2 receptors, increase protein synthesis via the mTOR-ERK pathway, resulting in dephosphorylation of eEF2 (*David et al., 2020*).

Consistent with our previous results that burst-induced potentiation in the presence of dopamine requires postsynaptic PKA and protein synthesis (*Fuchsberger et al., 2022*), we found here that synaptic potentiation induced by low frequency synaptic stimulation in the presence of dopamine following a priming protocol, which would otherwise induce synaptic depression (*Brzosko et al., 2015*), also requires postsynaptic PKA and new protein synthesis. While conventional t-LTP remained intact, blocking protein synthesis in the postsynaptic neurons completely prevented DA-LTP, suggesting that dopamine induces the synthesis of plasticity-related proteins required for converting synaptic depression into potentiation.

DA-LTP shares properties with L-LTP, which also requires dopamine signalling (*Frey et al., 1990*), PKA (*Frey et al., 1993*), and protein synthesis (*Frey et al., 1988*), and it was reported that protein synthesis was required hours after LTP induction for the maintenance of synaptic strength (*Frey et al., 1988*). In contrast, for the form of dopamine-dependent LTP studied here, protein synthesis was required ahead of or within the first few minutes of the induction protocol. Moreover, while previous studies have used extracellular application of protein synthesis inhibitors, here we loaded the protein synthesis inhibitors into the postsynaptic neuron via the patch pipette, suggesting that it is specifically postsynaptic protein synthesis that is required for DA-LTP.

We found that neuronal activity was also required for dopamine to increase protein synthesis. Moreover, we identified the $Ca^{2+}$-dependent AC subtypes AC1/AC8 to be involved in the induction of DA-LTP. This coincidence detector is stimulated by Gs-coupled dopamine D1/D5 receptor activation together with $Ca^{2+}$ influx (*Wayman et al., 1994*; *Ferguson and Storm, 2004*). Our results show that the AC1/AC8 subtypes and PKA are required for DA-LTP. These results suggest that dopamine application during neuronal activity induces the synthesis of plasticity-related proteins that enable synaptic potentiation.

Western blots showed that dopamine increased the levels of the GluA1 AMPA receptor subunit in a protein-synthesis-dependent manner, but that the GluA2 subunit remained unchanged. This is consistent with a previous study in hippocampal primary cultured neurons which reported that dopaminergic signalling increases the surface expression of AMPA receptor subunit GluA1 (*Smith et al., 2005*). The selective increase in the GluA1 over GluA2 subunit is interesting because of the important role ascribed to the GluA1 subunit in LTP (*Huganir and Nicoll, 2013*; *Diering and Huganir, 2018*).

We report here that although conventional t-LTP could still be induced in a GluA1 KO mouse model, GluA1 was required for DA-LTP. These observations revealed a possible mechanism through which dopamine can modulate synaptic strength. Furthermore, the findings indicated another intriguing possibility, namely that GluA1 homomeric AMPA receptors, which are calcium-permeable, mediate the expression of DA-LTP. CP-AMPA receptors appear to play a role in some forms of LTP (*Plant et al., 2006*; *Guire et al., 2008*; *Frey et al., 2009*; *Wakazono et al., 2024*), while other studies found no involvement (*Adesnik and Nicoll, 2007*; *Gray et al., 2007*).

The role of CP-AMPARs in synaptic plasticity remains controversial. We report here that they are required for DA-LTP, but not for conventional t-LTP. Importantly, even blocking CP-AMPARs after the pairing paradigm reversed DA-LTP, which supports the possibility that they are required for expression of DA-LTP. Whether they also contribute to the induction of the dopamine-dependent potentiation induced by low-frequency afferent stimulation following a t-LTD priming protocol (*Brzosko et al., 2015*) remains unresolved. Furthermore, it should be noted that using western blotting, we observed an increase in total GluA1 levels, but we cannot conclude whether DA increases surface GluA1 or CP-AMPA receptors from our experiments. While we show a requirement for CP-AMPA receptors for DA-LTP, it remains to be confirmed whether the pairing protocol together with dopamine indeed induces an increase in CP-AMPA receptors at the synapse. There are currently no suitable imaging techniques available to unambiguously quantify specific AMPA receptor subunit compositions at the synapse in acute slices.

Interestingly, previous studies have reported two forms of NMDA-receptor-dependent LTP, induced by spaced and compressed theta burst stimuli (TBS), that can be distinguished at hippocampal CA1 synapses based on their dependence on PKA (*Park et al., 2016*). Additionally, it was shown that the PKA-dependent form of LTP triggered by spaced stimulation leads to a transient increase in single-channel conductance, probably mediated by the insertion of calcium-permeable

(CP)-AMPA receptors (*Park et al., 2021*). The signalling pathway identified in our study overlaps with this pathway and, together, these findings suggest that two mechanistically distinct forms of LTP coexist at these synapses. It is possible that plasticity protocols that use extracellular electrodes also trigger the release of dopamine, and it would be of interest to investigate whether LTP induced by spaced TBS requires dopamine receptor activation and protein synthesis. In summary, our findings suggest a possible mechanism for how the reward-related neuromodulator dopamine may contribute to protein-synthesis-dependent synaptic plasticity facilitating hippocampal long-term memory. Signalling mechanisms underlying dopaminergic control of protein synthesis and synaptic weights may also be important for pathophysiological processes. Dysregulation in dopamine systems has long been implicated in drug addiction (for review, see *Dalley and Everitt, 2009*), schizophrenia (for review, see *Kahn et al., 2015*), and neurodegenerative diseases such as Parkinson's disease (for review, see *Poewe et al., 2017*), and, more recently, also Alzheimer's disease (*Nobili et al., 2017*). Understanding signalling pathways underlying dopamine-dependent synaptic plasticity may help guide future research into additional treatment options.

## Materials and methods

### Animals

Mice used for this study were housed at the Combined Animal Facility, Cambridge University. They were held on a 12 hr light/dark cycle at 19–23°C and were provided with water and food ad libitum. Experiments were carried out using wildtype C57BL/6 J mice (Charles River Laboratories, UK), and the transgenic mouse lines AC1 cyclase (AC) subtypes 1 and 8 double knockout (AC1/AC8 DKO) mice and GluA1 knockout (GluA1 KO) mice. The AC1/AC8 DKO mice have the genes for both AC1 and AC8 deleted globally. The mouse line was generated as described previously (*Wang et al., 2003*) and was imported from Michigan State University, MI, USA. The GluA1 KO mouse line was generated as described previously (*Zamanillo et al., 1999*) and was imported from the MRC Laboratory of Molecular Biology, Cambridge, UK.

All procedures were performed in accordance with the animal care guidelines of the UK Home Office regulations of the UK Animals (Scientific Procedures) Act 1986, and Amendment Regulations 2012, following ethical review by the University of Cambridge Animal Welfare and Ethical Review Body (AWERB). Animal procedures were authorised under Personal and Project licences held by the authors.

### Preparation of acute hippocampal slices

Male and female mice at postnatal days 12–19 were briefly anaesthetised with isoflurane (4% isoflurane in oxygen) and decapitated. The brain was rapidly removed into ice-cold artificial cerebrospinal fluid (aCSF; 10 mM glucose, 26.4 mM $NaH_2CO_3$, 126 mM NaCl, 1.25 mM $NaH_2PO_4$, 3 mM KCl, 2 mM $MgSO_4$, 2 mM $CaCl_2$) bubbled with carbogen gas (95% $O_2$/5% $CO_2$; pH 7.2, 270–290 mOsm/L) and glued to a vibrating microtome stage. Horizontal slices (350 µm) were sectioned with a vibratome (Leica VT 1200 S, Leica Biosystems, Wetzlar, Germany) and were subsequently submerged in aCSF for at least 1 hr at room temperature (RT) in a storage chamber. Slices were used 1–6 hr after preparation.

### Electrophysiology

#### Whole-cell recordings and synaptic plasticity

Individual slices were transferred to a submersion-type chamber for recordings and superfused with aCSF (2 ml/min) at RT (24–26 °C). Neurons were visualised with an infrared differential interference contrast (DIC) microscope using a ×40 water-immersion objective. Hippocampal subfields were identified and whole-cell patch-clamp recordings were performed on CA1 pyramidal neurons. Monopolar stimulation electrodes were placed in stratum radiatum for stimulation of Schaffer collaterals. Electrodes for test and control pathways were placed at the same distance (>100 µm) from either side of the recorded neuron. Patch pipettes (pipette resistance 4–7 MΩ) were pulled from borosilicate glass capillaries (0.68 mm inner diameter, 1.2 mm outer diameter) using a P-97 Flaming/Brown micropipette puller (Sutter Instruments Co., Novato, California, USA). Pipettes were filled with intracellular solution containing: 110 mM potassium gluconate, 4 mM NaCl, 40 mM HEPES, 2 mM ATP-Mg, 0.3 mM GTP (pH 7.2–7.3, 270–285 mOsm/L). The liquid junction potential was not corrected.

All experiments were performed in current-clamp mode. Cells were accepted for the experiment if their resting membrane potential was between −55 and −70 mV. Throughout the recording, the membrane potential was held at −70 mV by direct current application via the recording electrode. All cells were tested for regular spiking responses to positive current steps (20 pA, 800ms) characteristic of pyramidal neurons before the start of each recording.

TTX experiments were performed measuring spontaneous EPSPs for 5 min before and after adding 1 µM TTX to superfusing aCSF for 15 min.

Plasticity recordings were carried out as described previously (**Brzosko et al., 2015**). Briefly, EPSPs were evoked alternately in two input pathways (test and control) by direct current pulses at 0.2 Hz (stimulus duration 50 µs) through metal stimulation electrodes. The stimulation intensity was adjusted (100 µA – 500 µA) to evoke an EPSP with peak amplitude between 3 and 8 mV. After a stable EPSP baseline period of at least 10 min, spike-timing-dependent plasticity (STDP) was induced in the test pathway by repeated pairings (100 times at 0.2 Hz) of single evoked EPSPs and single postsynaptic action potentials elicited with the minimum somatic current pulse (1–1.8 nA, 3ms) via the recording electrode. Spike-timing intervals (Δt in ms) were measured between the onset of the EPSP and the onset of the action potential. Alternate stimulation of test and control EPSPs was resumed immediately after the pairing protocol and monitored for at least 40 min, except when the burst stimulation protocol was used for plasticity induction. In that case, stimulation of EPSPs was not resumed for an additional 10 min in the test pathway, and at the end of that period, five bursts, each of five action potentials at 50 Hz, were elicited with an inter-burst interval of 0.1 Hz by somatic current pulses (1.8 nA, 10ms) via the recording electrode. Immediately after the bursts, stimulation of EPSPs was resumed and monitored for at least 30 min. Presynaptic stimulation frequency to evoke EPSPs remained constant throughout the experiment. The unpaired control pathway served as a stability control.

## Drug application

Drugs were bath-applied to the whole slice through the perfusion system by dilution of concentrated stock solutions in aCSF, or by adding the drugs to the patch pipette solution when it was applied intracellularly to the postsynaptic cell only. For each set of recordings, interleaved control and drug conditions were carried out and were pseudo-randomly chosen. The following drugs were used in this study: 100 µM dopamine hydrochloride (Sigma-Aldrich, Dorset, United Kingdom), 10 µM CHX (Tocris Bioscience), 0.5 mM AM (stock solution in EtOH; Tocris Bioscience), 1 µM PKA inhibitor fragment (6-22) amide (Tocris Bioscience), 100 µM 1-naphthyl acetyl spermine trihydrochloride (NASPM trihydrochloride; Tocris Bioscience), 10 µM N,N,H,-trimethyl-5-[(tricyclo[3.3.1.13,7]dec-1-ylmethyl) amino]–1-pentanaminiumbromide hydrobromide (IEM1460; Tocris Bioscience).

## Data acquisition and data analysis of slice recordings

Data were collected using an Axon Multiclamp 700B amplifier (Molecular Devices, Sunnyvale, California, USA). Data were filtered at 2 kHz and were acquired and digitised at 5 kHz using an Instrutech ITC-18 A/D interface board (Instrutech, Port Washington, New York, USA) and custom-made acquisition procedures in Igor Pro (WaveMetrics, Lake Oswego, Oregon, USA).

All experiments were carried out in current clamp mode, and only cells with an initial series resistance from 9 to 16 MΩ were included. Series resistance was compensated for by adjusting the bridge balance, and recordings were discarded if series resistance changed by more than 30%. Offline analyses of plasticity recordings were done using custom-made procedures in Igor Pro. EPSP slopes were measured on the rising phase of the EPSP as a linear fit between the time points corresponding to 25–30% and 70–75% of the peak amplitude.

For quantifications, the mean EPSP slope per minute of the recording was calculated from 12 consecutive sweeps and normalised to the baseline (each data point in source data files is the mean of 12 averaged EPSPs). Normalised EPSP slopes from the last 5 mins of the baseline (immediately before pairing) and from the last 5 min of the recording were averaged. The magnitude of plasticity, as an indicator of change in synaptic weights, was defined as the average EPSP slope 40 min after the plasticity protocol expressed as a percentage of the average EPSP slope during baseline. For the burst-induction protocol, the change in synaptic weights was defined as the average EPSP slope 30 min after the plasticity protocol expressed as a percentage of the average EPSP slope during baseline.

## Statistical analysis of slice recordings

All data are presented as mean ± SEM. Statistical comparisons were performed using one-sample two-tailed, paired two-tailed, or unpaired two-tailed Student's $t$-test, with a significance level of $\alpha=0.05$. Significance level used was $\alpha=0.05$ and p values are indicated as *p<0.05, **p<0.01, ***p<0.001.

## Protein synthesis labelling in acute hippocampal slices

Incubation chambers were set up to submerge slices in oxygenated (95% $O_2$, 5% $CO_2$) aCSF containing a selection from the following drugs: 3 µM puromycin dihydrochloride (Sigma P8833), 100 µM dopamine hydrochloride (Sigma H8502), 10 µM SKF38393 (Sigma D047), 10 µM SCH23390 hydrochloride (Sigma D054), 50 µM sulpiride (Sigma S8010), 1 µM TTX (Tocris Bioscience 1078), 10 µM CHX (Tocris Bioscience 0970), 0.5 mM AM (stock solution in EtOH; Tocris Bioscience 1290/10), 30 µM Rp-cAMPS (Tocris Bioscience 1337).

Protein synthesis was measured using a PMY-based labelling assay adapted to acute hippocampal slices, similar to that described previously (*Schmidt et al., 2009*). Briefly, 3 µM PMY was used to incorporate into proteins during the elongation phase of synthesis (*Figure 1A*). After 30 min of incubation with PMY, samples were washed in PBS, and further processed for immunohistochemistry or western blotting as described below. The specificity of the conjugated PMY monoclonal antibody was confirmed with incubations of no-PMY controls and protein synthesis inhibitors.

## Immunohistochemistry

Immediately after incubations with PMY, slices were washed in PBS and fixed in 4% paraformaldehyde (PFA) in 1 x PBS for 24 hr at 4 °C. Slices were subsequently washed 3x5 min in PBST 0.01% (DPBS Thermo Fisher, Tween P1379 Sigma), permeabilised for 15 min in PBST 0.5% and washed in PBST 0.01% a further three times. Slices were incubated for 2 hr at RT in a blocking solution of 5% goat serum in PBST 0.0.1% (Sigma-Aldrich G9023) followed by an incubation overnight at 4 °C in blocking solution with anti-PMY antibody (clone 12D10, MABE343; 1:500) AlexaFluor 488 or AlexaFluor 647 (MABE343-AF488 or MABE343-AF647; Merck). On the following day, slices were left on a shaker for 2 hr at RT, followed by 3x5 min washes in PBST 0.01%. Finally, slices were mounted onto microscope slides (Thermo Fisher Scientific J1810AMNZ), allowed to dry for at least 2 hr at RT, then covered with Fluoroshield with Dapi mounting medium for nuclear staining (Sigma F6057) and sealed with coverslip sealant (Biotium 23005).

## Imaging and image analysis

Immunohistochemical preparations were visualised using a confocal laser-scanning microscope (Leica TCS SP8). Z-stacks of hippocampal CA1 region were taken using an HC PL APO 20 x/0.75 CS2 objective using the same exposure settings for all conditions that were compared to each other. AlexaFluor 647 was excited at 638 nm and emission detected at 671 nm; AlexaFluor488 was excited at 495 nm and emission detected at 519 nm, Dapi was excited at 359 nm, emission detected at 461 nm. Images were analysed using ImageJ. To obtain normalised integrated density, the maximum intensity Z-projection of slices were extracted, from which the integrated density was measured and normalised to the corresponding nuclear staining (Dapi) Z-projection measurement. Normalised integrated density was plotted and all data are presented as mean ± SEM. Statistical analysis was performed using one-way ANOVA and post-hoc Tukey's HSD test, following adherence to tests for normality and equality of variance. Significance level used was $\alpha=0.05$ and p values were indicated as *p<0.05, **p<0.01, ***p<0.001.

## Western blot analysis

Acute hippocampal slices were prepared and incubated with various compounds as described above. After 30 min incubation, slices were rapidly dissected in aCSF to obtain the CA1 region, which were pooled for slices from each condition and flash frozen in liquid N2. Protein extraction buffer (150 mM NaCl, 1% Triton x-100, 50 mM TrisHCl pH 7.4 with protease inhibitor [11836170001 Roche]), at a volume adjusted to the tissue weight for each sample, was added to lyse the tissue for 1 hr at 4 °C with frequent vortexing. Samples were centrifuged at 16,000 × $g$ for 30 min at 4 °C and the supernatant was retained. Lysate was mixed with LDS sample buffer (Invitrogen NP0007) and boiled at 95 °C

for 5 min, after which it was loaded onto a polyacrylamide gel (Bolt 4–12% Bis-Tris NW04127BOX). Loading was adjusted to achieve comparable actin signal. Proteins were transferred onto a nitrocellulose membrane using Trans-Blot Turbo Transfer (BioRad 1704158) and membranes were then blocked for 30 min at RT with 5% milk in TBST. Membranes were incubated with primary antibodies (anti-actin MAB1501R, anti-puromycin MABE343, anti-GluA1 AB1504, and anti-GluA2 AF1050), washed thrice in TBST, incubated with horseradish peroxidase (HRP)-coupled secondary antibody (Sigma) and washed a further three times. Membranes were developed using enhanced chemiluminescence and imaged with a ChemiDoc MP Imaging System (Bio-Rad). Quantification of antibody staining was made using ImageJ. Integrated density of GluA1/2 bands was normalised to integrated density of actin in the corresponding lane. All data are presented as mean ± SEM. Statistical analysis was performed using one-way ANOVA and post-hoc Tukey's HSD test, following adherence to tests for normality and equality of variance. Significance level used was $\alpha=0.05$ and p values were indicated as *$p<0.05$, **$p<0.01$, ***$p<0.001$.

## Acknowledgements

This research was supported by the Biotechnology and Biological Sciences Research Council, U.K. We are grateful for discussions of this project with other members of the Neuronal Oscillations Group.

## Additional information

### Funding

| Funder | Grant reference number | Author |
|---|---|---|
| Biotechnology and Biological Sciences Research Council | BB/V014641/1 | Tanja Fuchsberger Ole Paulsen |
| Medical Research Council | MC_U105174197 | Imogen Stockwell Ingo H Greger |
| Biotechnology and Biological Sciences Research Council | BB/N002113/1 | Imogen Stockwell Ingo H Greger |
| Wellcome Trust | 10.35802/223194 | Imogen Stockwell Ingo H Greger |

The funders had no role in study design, data collection and interpretation, or the decision to submit the work for publication. For the purpose of Open Access, the authors have applied a CC BY public copyright license to any Author Accepted Manuscript version arising from this submission.

### Author contributions

Tanja Fuchsberger, Conceptualization, Data curation, Formal analysis, Supervision, Funding acquisition, Validation, Investigation, Visualization, Methodology, Writing - original draft, Project administration; Imogen Stockwell, Data curation, Formal analysis, Investigation, Visualization, Writing - review and editing; Matty Woods, Data curation, Formal analysis, Investigation, Writing - review and editing; Zuzanna Brzosko, Investigation; Ingo H Greger, Resources, Funding acquisition, Writing - review and editing; Ole Paulsen, Conceptualization, Resources, Supervision, Funding acquisition, Project administration, Writing - review and editing

### Author ORCIDs

Tanja Fuchsberger https://orcid.org/0000-0002-4751-8806
Imogen Stockwell http://orcid.org/0000-0002-9712-1700
Ingo H Greger https://orcid.org/0000-0002-7291-2581
Ole Paulsen https://orcid.org/0000-0002-2258-5455

### Ethics

All procedures were performed in accordance with the animal care guidelines of the UK Home Office regulations of the UK Animals (Scientific Procedures) Act 1986, and Amendment Regulations 2012,

following ethical review by the University of Cambridge Animal Welfare and Ethical Review Body (AWERB). Animal procedures were authorized under Personal and Project licences (PP0499088) held by the authors.

Reviewer #1 (Public review): https://doi.org/10.7554/eLife.100822.3.sa1
Reviewer #2 (Public review): https://doi.org/10.7554/eLife.100822.3.sa2
Reviewer #3 (Public review): https://doi.org/10.7554/eLife.100822.3.sa3
Author response https://doi.org/10.7554/eLife.100822.3.sa4

---

## Additional files

### Supplementary files
MDAR checklist

### Data availability
Source data files have been provided for all Figures.

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
