## [Editor Report · eLife Assessment]

This manuscript addresses a mechanism by which dopamine (DA) regulates synaptic plasticity. The authors build upon their previous finding that DA applied after a timing pattern that ordinarily induces long-term depression (LTD) now induces long-term potentiation (LTP). The new findings that this ‘DA-dependent LTP’ involves de novo protein synthesis, a cyclicAMP signalling pathway, and calcium-permeable AMPA receptors (CP-AMPARs) are of **valuable** significance. The conclusions are **convincing** and largely supported by the evidence provided.

---

## [Referee Report · Reviewer #1 (Public review)]

Summary:

In this manuscript, Fuchsberger et al. demonstrate a set of experiments which ultimately identifies the de novo synthesis of GluA1-, but not GluA2-containing Ca2+ permeable AMPA receptors as a key driver of dopamine-dependent LTP (DA-LTP) during conventional post-before-pre spike-timing dependent (t-LTD) induction. The authors further identify adenylate cyclase 1/8, cAMP, and PKA as the crucial mitigators of these actions. While some comments have been identified below, the experiments presented are thorough and address the aims of the manuscript, figures are presented clearly (with minor comments), and experimental samples sizes and statistical analyses are suitable. Suitable controls have been utilized to confirm the role of Ca2+ permeable AMPAR. This work provides a valuable step forward built on convincing data towards understanding the underlying mechanisms of spike-timing dependent plasticity and dopamine.

Strengths:

Appropriate controls were used.

The flow of data presented is logical and easy to follow.

The quality of the data is solid.

Weaknesses:

Our concerns raised within the first round of review have been appropriately addressed by the authors.

---

## [Referee Report · Reviewer #2 (Public review)]

Summary:

The aim was to identify the mechanisms that underlie a form of long-term potentiation (LTP) that requires activation of dopamine (DA).

Strengths:

The authors have provided multiple lines of evidence that supports their conclusions; namely that this pathway involves activation of a cAMP / PKA pathway that leads to the insertion of calcium permeable AMPA receptors.

Weaknesses:

Some of the experiments could have been conducted in a more convincing manner.

---

## [Referee Report · Reviewer #3 (Public review)]

The manuscript of Fuchsberger et al. investigates the cellular mechanisms underlying dopamine-dependent long-term potentiation (DA-LTP) in mouse hippocampal CA1 neurons. The authors conducted a series of experiments to measure the effect of dopamine on the protein synthesis rate in hippocampal neurons and its role in enabling DA-LTP. The key results indicate that protein synthesis is increased in response to dopamine and neuronal activity in the pyramidal neurons of the CA1 hippocampal area, mediated via the activation of adenylate cyclases subtypes 1 and 8 (AC1/8) and the cAMP-dependent protein kinase (PKA) pathway. Additionally, the authors show that postsynaptic DA-induced increases in protein synthesis are required to express DA-LTP, while not required for conventional t-LTP.

The increased expression of the newly synthesized GluA1 receptor subunit in response to DA supports the formation of homomeric calcium-permeable AMPA receptors (CP-AMPARs). This evidence aligns well with data showing that DA-LTP expression requires the GluA1 AMPA subunit and CP-AMPARs, as DA-LTP is absent in the hippocampus of a GluA1 genetic knock-out mouse model.

Comments on revisions:

The authors addressed adequately all my comments.

---

## [Author Response]

The following is the authors’ response to the original reviews.

**Public Reviews:**

**Reviewer #1 (Public Review):**
Summary:In this manuscript, Fuchsberger et al. demonstrate a set of experiments that ultimately identifies the de novo synthesis of GluA1-, but not GluA2-containing Ca2+ permeable AMPA receptors as a key driver of dopamine-dependent LTP (DA-LTP) during conventional post-before-pre spike-timing dependent (t-LTD) induction. The authors further identify adenylate cyclase 1/8, cAMP, and PKA as the crucial mitigators of these actions. While some comments have been identified below, the experiments presented are thorough and address the aims of the manuscript, figures are presented clearly (with minor comments), and experimental sample sizes and statistical analyses are suitable. Suitable controls have been utilized to confirm the role of Ca2+ permeable AMPAR. This work provides a valuable step forward built on convincing data toward understanding the underlying mechanisms of spike-timing-dependent plasticity and dopamine.Strengths:Appropriate controls were used.The flow of data presented is logical and easy to follow.The quality of the data, except for a few minor issues, is solid.Weaknesses:The drug treatment duration of anisomycin is longer than the standard 30-45 minute duration (as is the 500uM vs 40uM concentration) typically used in the field. Given the toxicity of these kinds of drugs long term it's unclear why the authors used such a long and intense drug treatment.

In an initial set of control experiments (Figure S 1C-D) we wanted to ensure that protein synthesis was definitely blocked and therefore used a relatively high concentration of anisomycin and a relatively long pre-incubation period. We agree with the Reviewer that we cannot exclude the possibility that this treatment could compromise cell health in addition to the protein synthesis block. Therefore, we carried out an additional experiment with an alternative protein synthesis inhibitor cycloheximide at a lower standard concentration (10 µM) which confirmed a significant reduction in the puromycin signal (Figure S 1A-B). Together these results support the conclusion that puromycin signal is specific to protein synthesis in our labelling assay.

Furthermore, in the electrophysiology experiments, we used 500 μM anisomycin in the patch pipette solution. Under these conditions, we recorded a stable EPSP baseline for 60 minutes, indicating that the treatment did not cause toxic effects to the cell (Figure S1F). This high concentration would ensure an effective block of local translation at dendritic sites. Nevertheless, we also carried out this experiment with cycloheximide at a lower standard concentration (10 µM) and observed a similar result with both protein synthesis inhibitors (Figure 1F).

With some of the normalizations (such as those in S1) there are dramatic differences in the baseline "untreated" puromycin intensities - raising some questions about the overall health of slices used in the experiments.

We agree with the Reviewer that there is a large variability in the normalised puromycin signal which might be due to variability in the health of slices. However, we assume that the same variability would be present in the treated slices, which showed, despite the variability, a significant inhibition of protein synthesis. To avoid any bias by excluding slices with low puromycin signal in the control condition, we present the full dataset.

The large set of electrophysiology experiments carried out in our study (all recorded cells were evaluated for healthy resting membrane potential, action potential firing, and synaptic responses) confirmed that, generally, the vast majority of our slices were indeed healthy.

**Reviewer #2 (Public Review):**
Summary:The aim was to identify the mechanisms that underlie a form of long-term potentiation (LTP) that requires the activation of dopamine (DA).Strengths:The authors have provided multiple lines of evidence that support their conclusions; namely that this pathway involves the activation of a cAMP / PKA pathway that leads to the insertion of calcium-permeable AMPA receptors.Weaknesses:Some of the experiments could have been conducted in a more convincing manner.

We carried out additional control experiments and analyses to address the specific points that were raised.

**Reviewer #3 (Public Review):**
The manuscript of Fuchsberger et al. investigates the cellular mechanisms underlying dopamine-dependent long-term potentiation (DA-LTP) in mouse hippocampal CA1 neurons. The authors conducted a series of experiments to measure the effect of dopamine on the protein synthesis rate in hippocampal neurons and its role in enabling DA-LTP. The key results indicate that protein synthesis is increased in response to dopamine and neuronal activity in the pyramidal neurons of the CA1 hippocampal area, mediated via the activation of adenylate cyclases subtypes 1 and 8 (AC1/8) and the cAMP-dependent protein kinase (PKA) pathway. Additionally, the authors show that postsynaptic DA-induced increases in protein synthesis are required to express DA-LTP, while not required for conventional t-LTP.The increased expression of the newly synthesized GluA1 receptor subunit in response to DA supports the formation of homomeric calcium-permeable AMPA receptors (CP-AMPARs). This evidence aligns well with data showing that DA-LTP expression requires the GluA1 AMPA subunit and CP-AMPARs, as DA-LTP is absent in the hippocampus of a GluA1 genetic knock-out mouse model. Overall, the study is solid, and the evidence provided is compelling. The authors clearly and concisely explain the research objectives, methodologies, and findings. The study is scientifically robust, and the writing is engaging. The authors' conclusions and interpretation of the results are insightful and align well with the literature. The discussion effectively places the findings in a meaningful context, highlighting a possible mechanism for dopamine's role in the modulation of protein-synthesis-dependent hippocampal synaptic plasticity and its implications for the field. Although the study expands on previous works from the same laboratory, the findings are novel and provide valuable insights into the dynamics governing hippocampal synaptic plasticity.The claim that GluA1 homomeric CP-AMPA receptors mediate the expression of DA-LTP is fascinating, and although the electrophysiology data on GluA1 knock-out mice are convincing, more evidence is needed to support this hypothesis. Western blotting provides useful information on the expression level of GluA1, which is not necessarily associated with cell surface expression of GluA1 and therefore CP-AMPARs. Validating this hypothesis by localizing the protein using immunofluorescence and confocal microscopy detection could strengthen the claim. The authors should briefly discuss the limitations of the study.

Although it would be possible to quantify the surface expression of GluA1 using immunofluorescence, it would not be possible to distinguish between GluA1 homomers and GluA1-containing heteromers. It would therefore not be informative as to whether these are indeed CP-AMPARs. This is an interesting problem, which we have briefly discussed in the Discussion section.

Additional comments to address:(1) In Figure 2A, the representative image with PMY alone shows a very weak PMY signal. Consequently, the image with TTX alone seems to potentiate the PMY signal, suggesting a counterintuitive increase in protein synthesis.

We agree with the Reviewer that the original image was not representative and have replaced it with a more representative image.

(2) In Figures 3A-B, the Western blotting representative images have poor quality, especially regarding GluA1 and α-actin in Figure 3A. The quantification graph (Figure 3B) raises some concerns about a potential outlier in both the DA alone and DA+CHX groups. The authors should consider running a statistical test to detect outlier data. Full blot images, including ladder lines, should be added to the supplementary data.

We have replaced the western blot image in Figure 3A and have also presented full blot images including ladder lines in supplementary Figure S3.

Using the ROUT method (Q=1%) we identified one outlier in the DA+CHX group of the western blot quantification. The quantification for this blot was then removed from the dataset and the experiment was repeated to ensure a sufficient number of repeats.

**Recommendations for the authors:**

**Reviewer #1 (Recommendations For The Authors):**
(1) How the authors perform these experiments with puromycin, these are puromycilation experiments - not SuNSET. The SuNSET protocol (surface sensing of translation) specifically refers to the detection of newly synthesized proteins externally at the plasma membrane. I'd advise to update the terminology used.

We thank the Reviewer for pointing this out. We have updated this to ‘puromycin-based labelling assay’.

(2) The legend presented in Figure 2F suggests WT is green and ACKO is orange, however, in Figure 2G the WT LTP trace is orange, consider changing this to green for consistency.

We thank the Reviewer for this suggestion and agree that a matching colour scheme makes the Figure clearer. This has been updated.

(3) In the results section, it is recommended to include units for the values presented at the first instance and only again when the units change thereafter.

The units of the electrophysiology data were [%], this is included in the Results section. Results of western blots and IHC images were presented as [a.u.]. While we included this in the Figures, we have not specifically added this to the text of individual results.

(4) Two hours pre-treatment with anisomycin vs 30 minutes pretreatment with cycloheximide seems hard to directly compare - as the pharmokinetics of translational inhibition should be similar for both drugs. What was the rationale for the extremely long anisomycin pretreatment? What controls were taken to assess slice health either prior to or following fixation? This is relevant to the below point (5).

In an initial set of control experiments (Figure S 1C-D) we wanted to ensure that protein synthesis was definitely blocked and therefore used a relatively high concentration of anisomycin and a relatively long pre-incubation period. We agree with the Reviewer that we cannot exclude the possibility that this treatment could compromise cell health in addition to the protein synthesis block. Therefore, we carried out an additional experiment with an alternative protein synthesis inhibitor cycloheximide at a lower standard concentration (10 µM) which confirmed a significant reduction in the puromycin signal (Figure S1A-B). Together these results support the conclusion that puromycin signal is specific to protein synthesis in our labelling assay.

IHC slices were visually assessed for health. The large set of electrophysiology experiments carried out in our study (all recorded cells were evaluated for healthy resting membrane potential, action potential firing, and synaptic responses) also confirmed that, generally, the vast majority of our slices were indeed healthy.

(5) In Supplementary Figure 1, there is a dramatic difference in the a.u. intensities across CHX (B) and AM (D), please explain the reason for this. It is understood these are normalised values to nuclear staining, please clarify if this is a nuclear area.

We agree with the Reviewer that there is a large variability in normalised puromycin signal which may be due to variability in the health of the slices. However, we assume that the same variability would be present in the treated slices, which showed, despite the variability, a significant effect of protein synthesis inhibition. To prevent introducing bias by excluding slices with low puromycin signal in the control condition, we present the full dataset.

The CA1 region of the hippocampus contains of a dense layer of neuronal somata (pyramidal cell layer). We normalized against the nuclear area as it provides a reliable estimate of the number of neurons present in the image. This approach minimizes bias by accounting for variation in the number of neurons within the visual field, ensuring consistency and accuracy in our analysis.

(6) Please clarify the decision to average both the last 5 minutes of baseline recordings and the last 5 minutes of the recording for the normalisation of EPSP slopes.

The baseline usually stabilises after a few minutes of recording, thus the last 5 minutes were used for baseline measurement, which are the most relevant datapoints to compare synaptic weight change to. After induction of STDP, potentiation or depression of synaptic weights develops gradually. Based on previous results, evaluating the EPSP slopes at 30-40 minutes after the induction protocol gives a reliable estimate of the amount of plasticity.

**Reviewer #2 (Recommendations For The Authors):**
The concentration of anisomycin used (0.5 mM) is very high.

As described above, in an initial set of control experiments (Figure S 1C-D) we wanted to ensure that protein synthesis was definitely blocked and therefore used a relatively high concentration of anisomycin and a relatively long pre-incubation period. We agree with the Reviewer that this is higher than the standard concentration used for this drug and we cannot exclude the possibility that this treatment could compromise cell health in addition to the protein synthesis block. Therefore, we carried out an additional experiment with an alternative protein synthesis inhibitor cycloheximide at a lower standard concentration (10 µM) which confirmed a significant reduction in the puromycin signal (Figure S1A-B). Together these results support the conclusion that puromycin signal is specific to protein synthesis in our labelling assay.

Furthermore, in the electrophysiology experiments, we also used 500 µM anisomycin in the patch pipette solution. Under these conditions, we recorded a stable EPSP baseline for 60 minutes, indicating that the treatment did not cause toxic effects to the cell (Figure S1F). This high concentration would ensure an effective block of local translation at dendritic sites. Nevertheless, we also carried out this experiment with cycloheximide at a lower standard concentration (10 µM) and observed a similar result with both protein synthesis inhibitors (Figure 1F).

The authors conclude that the effect of DA is mediated via D1/5 receptors, which based on previous work seems likely. But they cannot conclude this from their current study which used a combination of a D1/D5 and a D2 antagonist.

We thank the Reviewer for pointing this out. We agree and have updated this in the Discussion section to ‘dopamine receptors’, without specifying subtypes.

There is no mention that I can see that the KO experiments were conducted in a blinded manner (which I believe should be standard practice). Did they verify the KOs using Westerns?

Only a subset of the experiments was conducted in a blinded manner. However, the results were collected by two independent experimenters, who both observed significant effects in KO mice compared to WTs (TF and ZB).

We received the DKO mice from a former collaborator, who verified expression levels of the KO mice (Wang et al., 2003). We verified DKO upon arrival in our facility using genotyping.

Maybe I'm misunderstanding but it appears to me that in Figure 1F there is LTP prior to the addition of DA. (The first point after pairing is already elevated). I think the control of pairing without DA should be added.

We thank the Reviewer for pointing this out. Based on previous results (Brzosko et al., 2015) we would expect potentiation to develop over time once DA is added after pairing, however, it indeed appears in the Figure here as if there was an immediate increase in synaptic weights after pairing. It should be noted, however, that when comparing the first 5 minutes after pairing to the baseline, this increase was not significant (t(9)=1.810, p = 0.1037). Nevertheless, we rechecked our data and noticed that this initial potentiation was biased by one cell with an increasing baseline, which had both the test and control pathway strongly elevated. We had mistakenly included this cell in the dataset, despite the unstable conditions (as stated in the Methods section, the unpaired control pathway served as a stability control). We apologise for the error and this has now been corrected (Figure 1F). In addition, we present the control pathway in Figure S1G and I.

We have also now included the control for post-before-pre pairing (Δt = -20 ms) without dopamine in a supplemental figure (Figure S1E and F).

The Westerns (Figure 3A) are fairly messy. Also, it is better to quantify with total protein. Surface biotinylation of GluA1 and GluA2 would be more informative.

We carried out more repeats of Western blots and have exchanged blots in Figure 3A.

We observed that DA increases protein synthesis, we therefore cannot exclude the possibility that application of DA could also affect total protein levels. Thus quantifying with total protein may not be the best choice here. Quantification with actin is standard practice.

While we agree with the Reviewer that surface biotinylation of GluA1 and GluA2 could in principle be more informative, we do not think it would work well in our experimental setup using acute slice preparation, as it strictly requires intact cells. Slicing generates damaged cells, which would take up the surface biotin reagents. This would cause unspecific biotinylation of the damaged cells, leading to a strong background signal in the assay.

In Figure 4 panels D and E the baselines are increasing substantially prior to induction. I appreciate that long stable baselines with timing-dependent plasticity may not be possible but it's hard to conclude what happened tens of minutes later when the baseline only appears stable for a minute or two. Panels A and B show that relatively stable baselines are achievable.

We agree with the Reviewer that the baselines are increasing, however, when looking at the baseline for 5 minutes prior to induction (5 last datapoints of the baseline), which is what we used for quantification, the baselines appeared stable. Unfortunately, longer baselines are not suitable for timing-dependent plasticity. In addition, all experiments were carried out with a control pathway which showed stable conditions throughout the recording.

In general, the discussion could be better integrated with the current literature. Their experiments are in line with a substantial body of literature that has identified two forms of LTP, based on these signalling cascades, using more conventional induction patterns.

We thank the Reviewer for this suggestion and have added more discussion of the two forms of LTP in the Discussion section.

It would be helpful to include the drug concentrations when first described in the results.

Drug concentration have now been included in the Results section.

It is now more common to include absolute t values (not just <0.05 etc).

While we indicate significance in Figures using asterisks when p values are below the indicated significance levels, we report absolute values of p and t values in the Results section.

Similarly full blots should be added to an appendix / made available.

We have now included full blot images in Supplementary Figure S3.

A 30% tolerance for series resistance seems generous to me. (10-20% would be more typical).

We thank the Reviewer for their suggestion, and will keep this in mind for future studies. However, the error introduced by the higher tolerance level is likely to be small and would not influence any of the qualitative conclusions of the manuscript.

Whereas series resistance is of course extremely important in voltage-clamp experiments, changes in series resistance would be less of a concern in current-clamp recordings of synaptic events. We use the amplifier as a voltage follower, and there are two problems with changes in the electrode, or access, resistance. First, there is the voltage drop across the electrode resistance. Clearly this error is zero if no current is injected and is also negligible for the currents we use in our experiments to maintain the membrane voltage at -70 mV. For example, the voltage drop would be 0.2 mV for 20 pA current through a typical 10 MOhm electrode resistance, and a change in resistance of 30% would give less than 0.1 mV voltage change even if the resistance were not compensated. The second problem is distortion of the EPSP shape due to the low-pass filtering properties of the electrode set up by the pipette capacitance and series resistance (RC). This can be a significant problem for fast events, such as action potentials, but less of a problem for the relatively slow EPSPs recorded in pyramidal cells. Nevertheless, we take on board the advice provided by the Reviewer and will use the conventional tolerance of 20% in future experiments.

**Reviewer #3 (Recommendations For The Authors):**
In the references, the entry for Burnashev N et al. has a different font size. Please ensure that all references are formatted consistently.

We thank the Reviewer for spotting this and have updated the font size of this reference.